# Fabrication of Complex Three-Dimensional Structures of Mica through Digital Light Processing-Based Additive Manufacturing

**Sinuo Zhang** [1,2], **Imam Akbar Sutejo** [1,2], **Jeehwan Kim** [2,3], **Yeong-Jin Choi** [2], **Chang Woo Gal** [2] **and Hui-suk Yun** [1,2,*]

1 Advanced Materials Engineering, University of Science and Technology (UST), Daejeon 34113, Korea
2 Department of Advanced Biomaterials Research, Ceramics Materials Division, Korea Institute of Materials Science (KIMS), Changwon-si 51508, Korea
3 School of Materials Science and Engineering, Pusan National University, Busan 46241, Korea
* Correspondence: yuni@kims.re.kr; Tel.: +82-55-280-3351; Fax: +82-55-280-3392

**Abstract:** Mica is a group of clay minerals that are frequently used to fabricate electrical and thermal insulators and as adsorbents for the treatment of cationic pollutants. However, conventional subtractive manufacturing has the drawback of poor three-dimensional (3D) shape control, which limits its application. In this study, we propose digital light processing (DLP)-based additive manufacturing (AM) as one of the most effective ways to address this drawback. Two major challenges for the ceramic DLP process are the production of a homogeneous and stable slurry with the required rheological properties and the maintenance of printing precision. The mica green body was fabricated using a 53 vol.% solid loading slurry through DLP, which exhibited good dimensional resolution under an exposure energy dose of 10 mJ/cm$^2$. The precise, complex 3D structure was maintained without any defects after debinding and sintering at 1000 °C. The use of ceramic AM to overcome the shape-control limitations of mica demonstrated in this study offers great potential for expanding the applications of mica.

**Keywords:** clay minerals; mica; additive manufacturing; digital light processing

## 1. Introduction

Mica is the general term for the mica group of clay minerals, which are aluminosilicates of metals, such as potassium, aluminum, magnesium, iron, and lithium. Mica is a key insulating raw material for constructing electrical equipment owing to its outstanding electrical insulation [1], thermal insulation properties [2,3], acid and alkali resistance [4], and compression resistance. In addition, mica has a layered structure with two layers of silicic acid tetrahedron sandwiching one layer of an aluminum–oxygen octahedron. The isomorphous substitution of Si/Al/Mg in the tetrahedron/octahedron by low-valent cations imparts a permanent negative charge to mica, making it an effective adsorbent for the treatment of cationic pollutants [5–7]. With the gradual expansion of the application field of mica, the demand for its structural control is also increasing. Insulators with three-dimensional (3D) structures are required for specific application scenarios. For adsorbents, increasing the complexity of the internal geometric structure is also an effective approach to increase the adsorption capacity [8]. However, the precise control of 3D structures with traditional manufacturing methods such as pressing and casting is still challenging.

Additive manufacturing (AM), also known as 3D printing, is expected to address the drawbacks of traditional manufacturing methods for precise structural control of mica. AM is a group of technologies in which a structure is fabricated directly from a virtual model by stacking material (rather than subtracting or forming) layer-by-layer. Several studies have investigated the possibility of using AM to extend the properties and applications of clay

minerals. Franchin et al. used metakaolin-based ink to fabricate a log-pile structure using direct ink writing (DIW) AM. After four cycles, the printed mineral sorbent still exhibited a high cation exchange capacity, with an $NH^{4+}$ removal efficiency of 80%, making it a viable alternative to manufactured zeolites and traditional ceramics [9]. Chan et al. investigated the effect of modifying the solid loading of aqueous clay suspensions using DIW on their printability, or material extrusion, and eventually 3D-printed complex decorative items for architectural purposes [10]. Nevertheless, the majority of research on AM of clay minerals now focuses on extrusion-based techniques, with publications on higher-precision AM techniques being few.

Ceramic stereolithography based on digital light processing (DLP) is one of the most promising AM technologies, owing to its high precision. In the ceramic DLP process, the photocurable slurry is subjected to ultraviolet (UV) light of a certain wavelength to build a ceramic green body layer-by-layer. Obtaining a homogeneous and stable slurry with the requisite rheological characteristics [11] and maintaining printing precision [12] are two major challenges associated with the ceramic DLP process. The rheological properties of the slurry can be controlled by adjusting the particle size, surface area, wettability as well as resin condition, etc. While the curability and printing accuracy of the slurry are affected by factors such as the color of the powder, the composition of the resin, and the light absorption of the slurry [13]. In our previous study, we explored the possibility of DLP printing with the natural mineral illite, focusing on the influence of the natural mineral's deep color on its slurry photocuring ability [14]. Mica fabrication using DLP-based AM has not yet been reported, possibly because mica is a naturally occurring material with a large particle size that makes preparing the required slurry difficult [15]. Additionally, because mica is generally deeper in color, it absorbs more light energy than white-colored ceramic powders, which may result in more difficult UV curing [16].

In this study, the possibility of manufacturing a 3D structure of mica using DLP-AM was investigated by optimizing the slurry preparation and printing parameters. Initially, the effects of particle size and distribution on the homogeneity and stability of mica slurry were investigated, followed by the effect of slurry composition on the rheological characteristics. The photocuring behavior of the optimal slurry was assessed using different exposure parameters to ensure printing precision. Mica ceramic 3D structures with complex geometries were constructed based on these findings, and their shrinkage rates, densities, and microstructures were examined after sintering at various temperatures.

## 2. Materials and Methods

Mica (INT Korea, Ansan, Korea) was utilized as a ceramic powder. DISPERBYK-180 (Altana AG, Wesel, Germany), a frequently used dispersing additive for stabilizing inorganic particles, was selected as the dispersing agent for producing the slurry. The photosensitive resin consisted of the combination of a trifunctional acrylate monomer, trimethylolpropane triacrylate (TMPTA, Sigma-Aldrich, St. Louis, MO, USA), and a low-viscosity bifunctional acrylate monomer, 1,6-hexanediol diacrylate (HDDA, Sigma-Aldrich, USA). As a photoinitiator, irgacure (BASF, Ludwigshafen, Germany) was employed.

The as-received mica powder was pulverized through planetary milling at 450 rpm for 2.5 h in ethanol and subsequently coated with a 3–7 wt.% DISPERBYK-180 by ball milling for 24 h. The dispersant concentration was determined from the minimum viscosity of the ceramic suspension against the dispersant content. TMPTA, HDDA, and the photoinitiator were combined in a planetary centrifugal mixer (THINKY, Laguna Hills, CA, USA) for 10 min at 2000 rpm to create the photocurable resin. Dispersant-coated mica powder was then added and mixed well for a further 5 min with a solid loading of 45–54 vol.%. By observing the rheological characteristics of the ceramic slurry as the powder content was increased, the quantity of solid loading was calculated.

SolidWorks software (SolidWorks Corp., Waltham, MA, USA) was used to create 3D models of mica. Using a 3D ceramic printing system that was created in-house, the ceramic pieces were manufactured using a UV light source with a wavelength of 405 nm. More

information about the printing system is provided in our previous work [14]. For all builds, the layer thickness was set to 20 μm, and the UV light energy was adjusted to 10 mW/cm$^2$. Using ethanol, uncured slurries were eliminated from the printed pieces.

The debinding profile was designed and optimized based on the results of thermogravimetric analysis (TGA), as shown in Figure 1. With a maximum temperature of 600 °C and a heating rate of 1 °C/min, TGA was carried out using a TGA55 analyzer (TA Instruments, New Castle, DE, USA). The specimens were first heated to 600 °C under flowing N$_2$ to remove any remaining organic materials; then they were consolidated by heating at 800, 900, 1000, and 1100 °C for 4 h, respectively, in a box furnace under air.

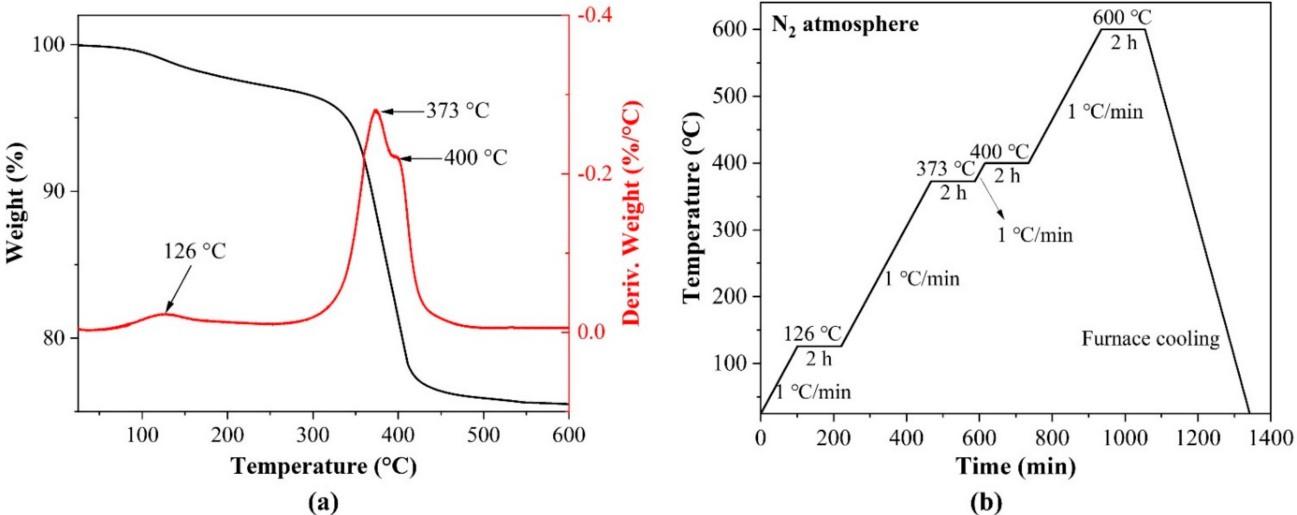

**Figure 1.** (**a**) TGA result of mica green body and (**b**) debinding profile.

To optimize slurry preparation, the rheological behaviors of several suspensions were examined using a rheometer (Discovery HR-1, TA Instruments, USA). As a function of shear rate in the range of 1–100 s$^{-1}$, the viscosity was tested under isothermal circumstances (35.0 °C).

The cure depth and excess cure width were established using the digital light projector of the 3D ceramic printing system to polymerize a thin layer of the slurry. Five sets of experiments were performed to investigate the effect of the exposure energy dose on the cure depth and excess cure width, as shown in Table 1. After polymerization, the cured layers were washed with ethanol to remove any remaining slurries. The thickness of the polymerized sheets was then measured using a micrometer (High-Accuracy Digimatic$^®$ Digital Micrometer; Mitutoyo, Kawasaki, Japan). The width of the blank region, which was measured using an optical microscope (SV-55, SOMETECH VISION, Seoul, Korea), was subtracted from the initial design width to get the excess cure width.

**Table 1.** Exposure parameters used for a series of experiments designed to control cure depth and excess cure width in which E = I × T.

| Light Exposure Time T (s) | Light Intensity I (mW/cm$^2$) | Exposure Energy Dose E (mJ/cm$^2$) |
|---|---|---|
| 1 | 10 | 10 |
| 2 | 10 | 20 |
| 3 | 10 | 30 |
| 4 | 10 | 40 |
| 5 | 10 | 50 |

To visualize the microstructures of the green and sintered samples, a field-emission scanning electron microscope (JSM-7500F FE-SEM, JEOL, Tokyo, Japan) was employed. The density and shrinkage rate of sintered samples were measured using a $10 \times 5 \times 2.5$ mm (length $\times$ width $\times$ height) cube ($n = 5$). The Archimedes method (ASTM C373-88) was used to compute the densities of the samples. X-ray diffraction (XRD, Dmax-2500, Rigaku, The Woodlands, TX, USA, 40 kV, 100 mA) was used to evaluate the chemical composition of the samples.

## 3. Results and Discussions

### *3.1. Effects of Particle Size Distribution on the Homogeneity and Stability of Mica Slurry*

Homogeneous and stable ceramic slurry feed is essential for the ceramic 3D printing process [17]. The homogeneity and stability of the ceramic slurry not only affect the continuous supply of feed, especially for film-type systems, but also the uniformity of the final fabricated specimen [18]. The particle size distribution of the powder is one of the major determinants of homogeneity and stability [19]. Therefore, controlling the particle size distribution of the powder to improve the printability of the ceramic slurry feed was investigated.

### 3.1.1. Characteristics of Mica Powder

As a naturally occurring clay mineral, mica typically has a relatively large particle size and wide distribution. Generally, in ceramic slurries, a larger particle size of the ceramic powder results in a higher possibility of the powder to sediment under gravity, resulting in the inhomogeneity of the slurry [20]. As a result, the as-received powder was planetary milled for 2.5 h to reduce the particle size and narrow the distribution, and the morphology and particle size distribution of the planetary milled mica powder is shown in Figure 2. Mica was laminated, as observed in the SEM images (Figure 2a), and the structural features of the powder did not change as a result of planetary milling; however, the particle size was significantly reduced. The particle size of the as-received mica powder was bimodally distributed (30–31 μm, 109–110 μm), with an average particle size of $d_{50} = 28.36$ μm. After planetary ball milling, the particle size exhibited a unimodal distribution, the peak bigger than 100 μm disappeared, and the particle size was reduced by nearly half ($d50 = 14.85$ μm). In addition, the XRD characteristic peaks of the powder were consistent with muscovite $2M_1$ (ICCD:00-046-1409) before and after planetary ball milling (Figure 2c), indicating that the crystal grains were not contaminated with any impurities during milling and the properties of the raw mica were preserved.

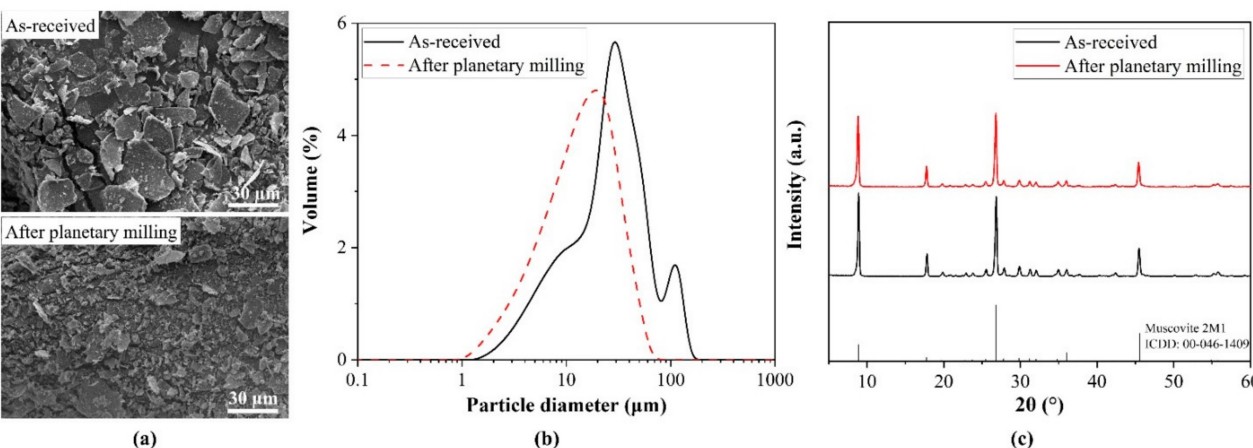

**Figure 2.** (**a**) Morphology, (**b**) particle size distribution, and (**c**) XRD patterns of the as-received and planetary milled mica powder.

### 3.1.2. Homogeneity and Stability of the Mica Slurry

To investigate whether the particle size and distribution affect the homogeneity and stability of the slurry, two types of slurries with the same resin formulation and solid loading were prepared from the as-received and planetary milled mica powder. Owing to the large particle size, preparing a slurry with a high solid loading using as-received mica powders is challenging. On the other hand, the following observation of sedimentation would be easier when there was a relatively lower solid loading; thus, the solid loading was fixed at 40 vol.%. The status of the two slurries in the chamber and after being processed by the doctor blade is shown in Figure 3a,b. Figure 3a shows that the as-received mica powder is not consistently dispersed in the slurry, which contributes to its poor wettability and prevents it from being uniformly spread across the film after being processed by the doctor blade. On the other hand, the slurry made from the planetary milled mica powder was considerably more homogeneous and effectively formed a thin, uniform layer after being processed by the doctor blade (Figure 3b). In addition, the stability of the two slurries was further investigated by standing the slurries at room temperature and observing sedimentation. Figure 3c,d show the sedimentation of the as-received and planetary milled mica slurries over time, respectively. Settling of the as-received mica slurry occurred quickly and clearly over time. The sedimentation volume ratio dropped to 94% after only 0.5 days and to 77% after 5 days. Because 3D printing is a time-consuming technology, if the slurry is not stable over a period of time, it will result in unevenness and imperfections in the printed specimen. In contrast, the slurry prepared from the powder with a reduced particle size by planetary milling was relatively stable. The slurry exhibited no visible signs of separation or sedimentation after 0.5 days of standing. The sedimentation volume ratio after 5 days was 95%, which was higher than that of as-received mica after 0.5 days (77%). The planetary milled mica slurry did not only generate a uniform thin layer after being processed by the doctor blade, but the sedimentation volume ratio was also significantly increased. As a result, the mica powder after planetary milling was utilized as the feed material in the following experiments to ensure the uniformity and precision of the printed specimens.

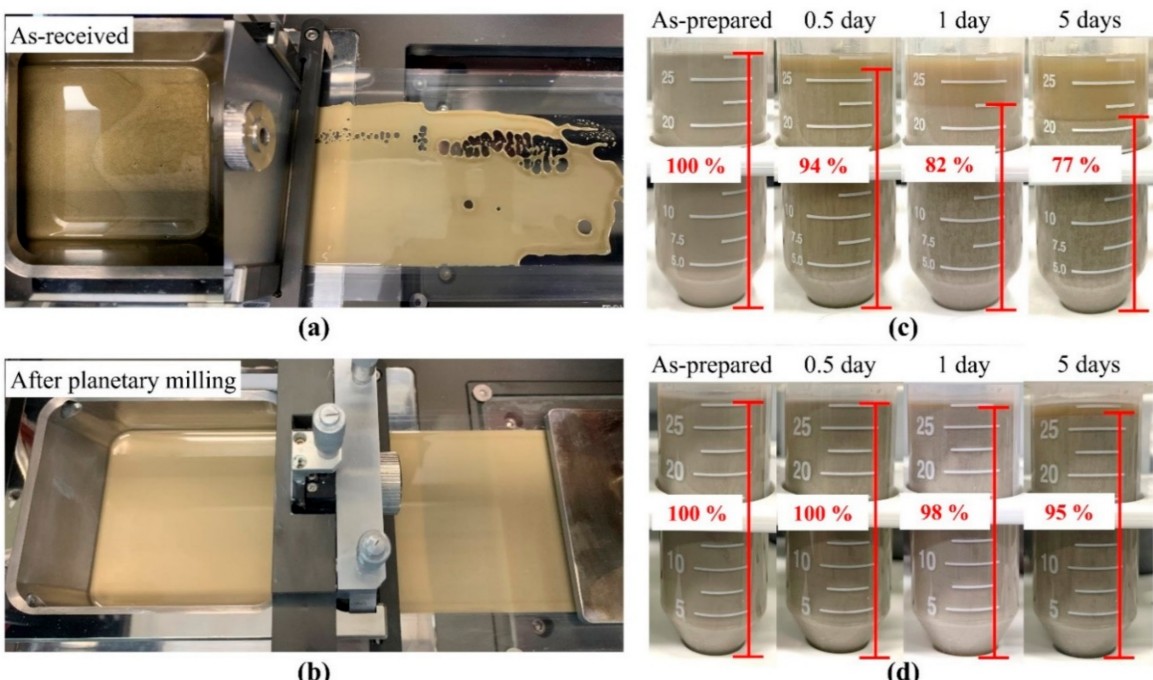

**Figure 3.** State of the (**a**) as-received mica slurry and (**b**) planetary milled mica slurry after passing through the doctor blade. Sedimentation of the (**c**) as-received and (**d**) planetary milled mica slurries (solid loading = 40 vol.%).

### 3.2. Rheological Behavior of Mica Slurry

The dispersant coating of ceramic powder is another effective approach to improving slurry stability and reducing viscosity [21]. HDDA is a hydrophobic resin monomer with outstanding comprehensive characteristics that are frequently used in preparing UV-curable ceramic slurries. On the other hand, mica powder has a water contact angle of less than 5°, making it exceptionally hydrophilic. The incompatibility of hydrophilic powders with hydrophobic HDDA results in particle agglomeration and sedimentation [22–24]. To mitigate the influence of this adverse element, a dispersant coating is required to change the wettability of mica powder.

The viscosity of the planetary milled mica slurry as a function of the dispersant concentration is shown in Figure 4a. When the dispersant concentration was low, the powder particles tended to agglomerate owing to Van der Waals forces [25]. The powder was not homogeneously dispersed in the resin and had a high viscosity. However, as the concentration of the dispersant increased, the viscosity of the slurry gradually decreased, and the viscosity reached the minimum at a concentration of 7 wt.%. This indicates that when the dispersant concentration was 7 wt.%, the dispersant coats the surface of the powder well, realizes the steric hindrance, and reduces the viscosity to the minimum. However, when the concentration was increased to 8 wt.%, the viscosity of the slurry slightly increased again. This was due to the excess dispersant creating flocculation, which damaged the stability of the slurry, thereby increasing the viscosity again [26]. As a consequence, the optimal concentration of the dispersant was 7 wt.%.

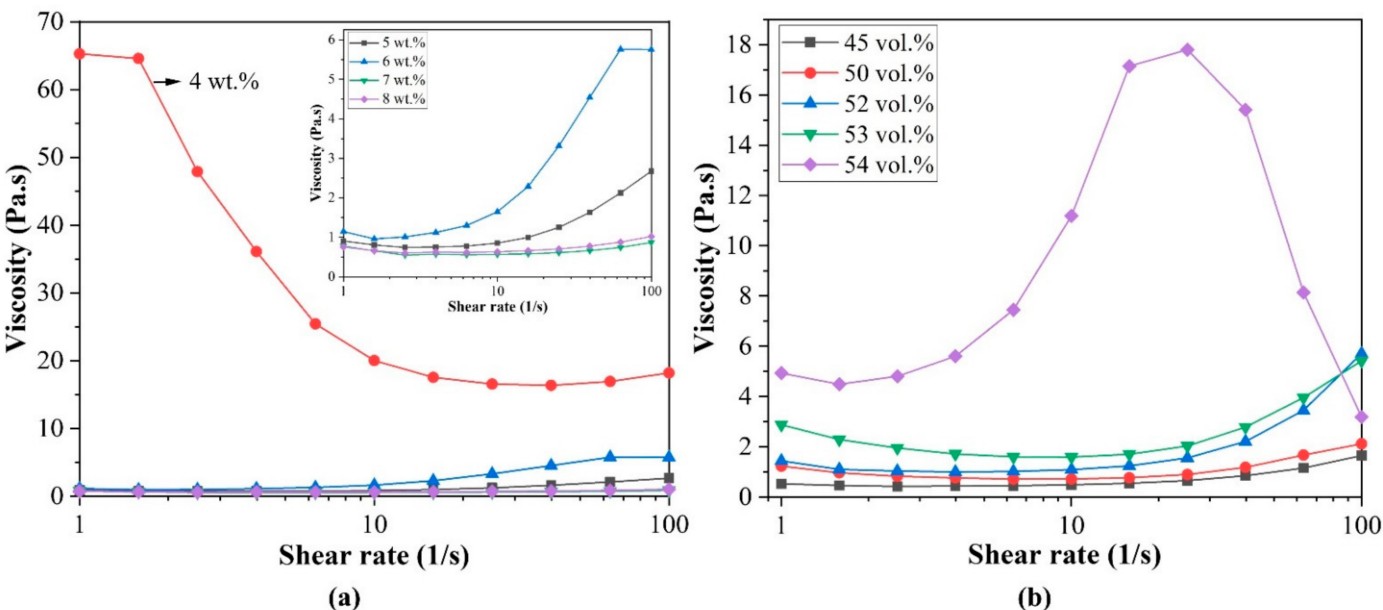

**Figure 4.** Viscosity curves of the planetary milled mica slurry with different (**a**) concentrations of the dispersant and (**b**) solid loading (dispersant concentration = 7 wt.%).

The relationship between solid loading and slurry rheological properties was further investigated using the optimal dispersant concentrations. Low solid loading causes a higher shrinkage rate; therefore, certain cracks and defects are unavoidable during debinding [27]. However, increasing the solid loading also increases the viscosity of the slurry and even affects its homogeneity and stability. Therefore, to increase the relative density after sintering and to obtain a more perfect specimen, increasing the solid loading of the slurry as much as possible is necessary to ensure uniformity, stability, and printability. Figure 4b shows the effect of slurry solid loading on viscosity. When the solid loading was increased to 54 vol.%, the viscosity of the slurry increased dramatically. Moreover, shear thickening occurred in the low shear rate region (1–10/s), which is not favorable for the DLP system [28]. As a result, the solid loading for the planetary milled mica slurry can

be increased to 53 vol.% to ensure slurry spreading and accurate printing. The planetary milled mica slurry was fixed at 7 wt.% dispersant and 53 vol.% solid loading.

### 3.3. Cure Behavior of the Mica Slurry

Upon irradiation of UV light on the curable slurry, the UV light of the resin is initiated, forming a cured layer that ensures the printability of the slurry. In addition, the inclusion of ceramic powder in the slurry is thought to produce light scattering owing to the difference in the refractive indices between the ceramic and photocurable resin, which can generate an excess cure width beyond the designed shape and affect printing accuracy [29,30]. The cure depth and excess cure width of the optimized mica slurry with 53 vol.% solid loading were tested to assure printability and geometric accuracy throughout the DLP process. According to The Beer–Lambert law, the cure depth ($C_d$) is a function of the exposure energy dose (E), depth critical energy dose ($E_d$), and depth sensitivity ($S_d$), as indicated in Equation (1) [23,31–33]

$$C_d = S_d \ln(\frac{E}{E_d}), \tag{1}$$

Following the Quasi-Beer–Lambert model, Equation (2) defines the excess cure width [23,31–33]

$$W_{ex} = S_w \ln(\frac{E}{E_w}), \tag{2}$$

$S_w$ stands for width sensitivity, and $E_w$ for width critical energy dose.

Figure 5a illustrates how the incident energy dose affects the curing behavior. The incident energy logarithm served as the independent variable, and the cure depth and width served as the dependent variables. The critical energy doses $E_w$ and $E_d$ can be calculated from the energy intercepts, and the attenuation factors $S_d$ and $S_w$ can be calculated from the slopes. It is clear that the results in Figure 5a concur with Equations (1) and (2). Table 2 contains the critical energy doses ($E_d$ and $E_w$) and sensitivities ($S_d$ and $S_w$). It can be observed that the width sensitivity is lower than the depth sensitivity, implying that the printing precision is easier to adjust. The depth critical energy dose represents the minimal exposure energy dose required for curing a specific slurry, while the width critical energy dose indicates that the light energy is larger than this value before the excess cure width is generated [31]. Table 2 shows that $E_w$ is more than three times that of $E_d$, and, as shown in Figure 5a, the excess cure width decreased dramatically as the exposure energy dose decreased. This indicates that less energy was available to cause scattering, resulting in a significant improvement in printing accuracy. Furthermore, Figure 5b shows that there was no noticeable excess curing when the exposure energy dose was 10 mJ·cm$^{-2}$. The geometry of the void shrunk as the exposure energy dose increased, implying that the over-curing induced by scattering became increasingly severe. Based on these findings, the thickness of the printing layer was set to 20 μm, and the exposure energy dose of 10 mJ·cm$^{-2}$ was chosen for all subsequent printing. As a result, a cure depth of approximately 66 μm was achieved, which is more than three times the layer thickness and should be sufficient to ensure interlayer adhesion.

**Table 2.** Sensitivities ($S_d$ and $S_w$) and critical energy doses ($E_d$ and $E_w$) of the mica slurry (solid loading = 53 vol.%).

| Solid Loading | $S_d$ (μm) | $S_w$ (μm) | $E_d$ (mJ/cm$^2$) | $E_w$ (mJ/cm$^2$) |
|---|---|---|---|---|
| 53 vol.% | 42.91 | 34.09 | 2.11 | 6.91 |

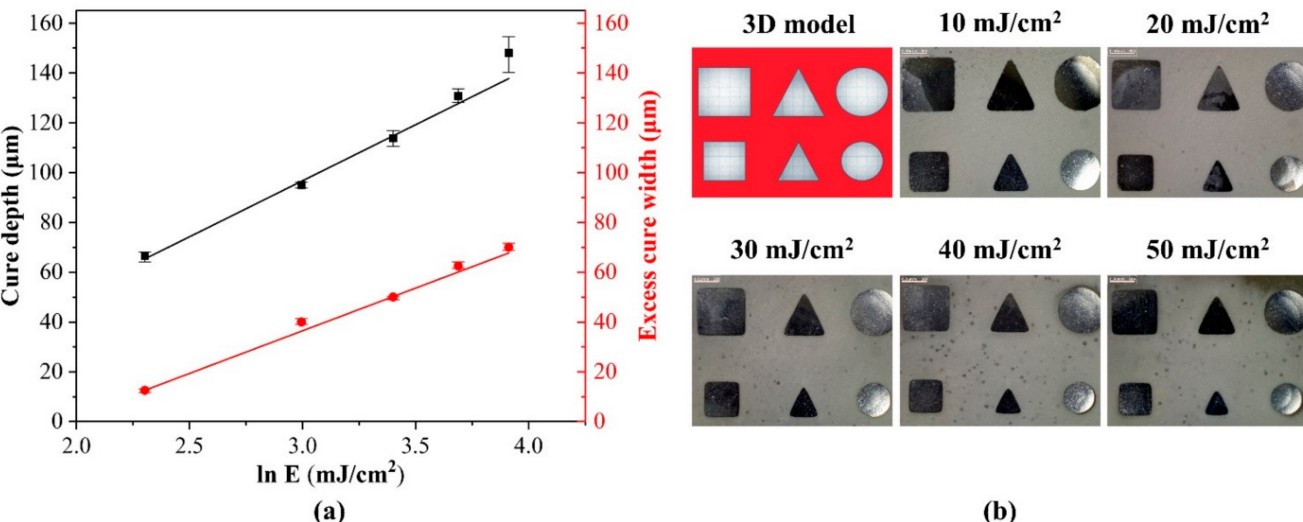

**Figure 5. (a)** Effects of exposure dose on the cure depth and excess cure width of the planetary milled mica slurry (53 vol.%). **(b)** Optical images of cured monolayers depending on the exposure dose.

### 3.4. Properties of the Structure of Mica Ceramics

Figure 6 shows the effects of the sintering temperature on the shrinkage rate and relative density of the sintered mica when the solid loading was set to 53 vol.%. As shown in Figure 6, the shrinkage rate and relative density increased with an increase in the sintering temperature. Moreover, the ceramic particles became more aggregated, and the ceramic became denser with increasing sintering temperature. When the sintering temperature reached 1100 °C, the linear shrinkage was 17.43%, and the relative density was up to 96.04%. Figure 7 shows FE-SEM images of the mica green body, debound body, and sintered body at different temperatures. As shown in Figure 7a, the microstructure of the DLP-fabricated mica green body was uniform and integrated after optimization of the slurry and manufacturing parameters. After debinding, the debound body (Figure 7b) became a porous, non-dense structure with the volatilization of the organic binding agent inside the green body. The sintered body became denser with increasing temperature, which is in good agreement with the relative density results shown in Figure 6. This demonstrates that the microstructure can be modified by adjusting the sintering temperature, thereby expanding the range of mica applications.

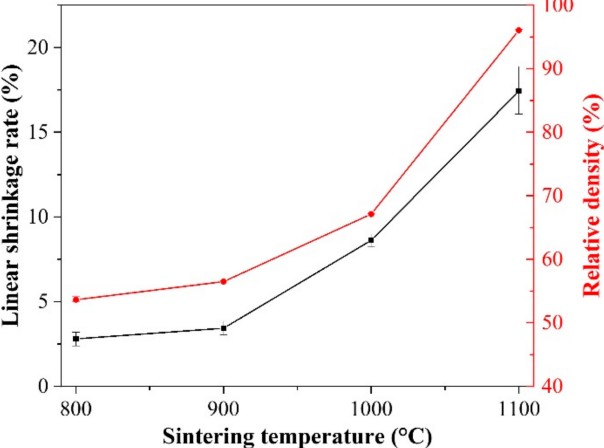

**Figure 6.** Shrinkage rate and relative density of the mica specimens sintered at different temperatures.

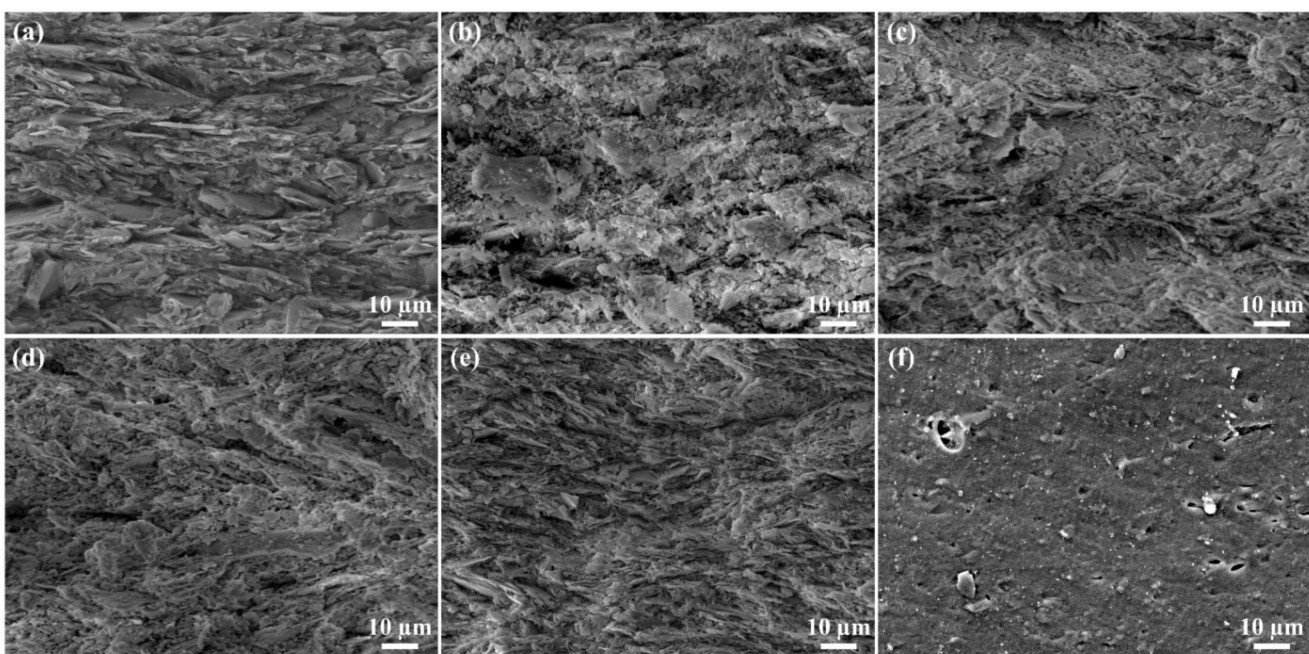

**Figure 7.** FE-SEM images of the microstructure of the (**a**) mica green body; (**b**) debound body; and sintered body at (**c**) 800 °C, (**d**) 900 °C, (**e**) 1000 °C, and (**f**) 1100 °C.

Figure 8 shows the XRD patterns of the mica ceramics sintered at different temperatures. Compared with Figure 2, the characteristic peaks of the sintered mica ceramics shifted slightly, and the original matching with muscovite $2M_1$ (ICCD:00-046-1409) became consistent with potassium mica (ICCD:00-046-0741). This was due to high-temperature sintering, which dehydroxylated the initial powder and caused a slight crystal form shift [34]. Furthermore, the intensity of the characteristic peaks gradually weakened as the sintering temperature increased, indicating that the lattice structure of mica was gradually destroyed. When the temperature was below 1100 °C, mica retained its lattice structure; however, the main characteristic peaks disappeared after the temperature of 1100 °C was reached, indicating that the crystal structure was destroyed. This might be because the mica powder used in this experiment reached its melting point at 1100 °C, and after sintering, the mica may recrystallize during the furnace cooling causing minor peaks to appear in the XRD spectrum. This observation is consistent with the results of FE-SEM (Figure 7).

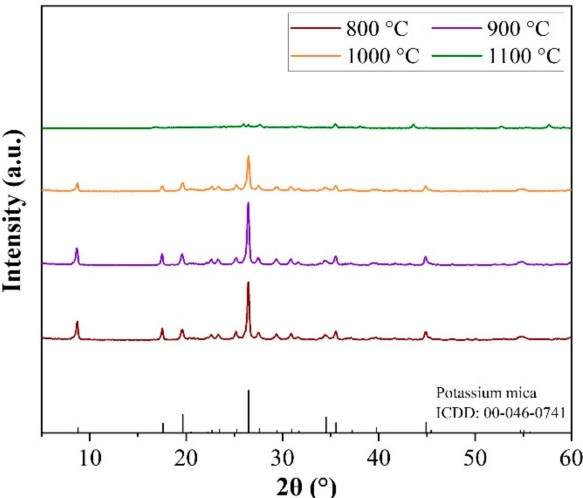

**Figure 8.** XRD patterns of mica ceramics sintered at different temperatures.

### 3.5. Fabrication of a 3D Complex Structure of Mica

Finally, with the optimization of the slurry preparation (53 vol.% solid loading) and printing parameters (10 mJ/cm$^2$ exposure energy dose, 20 µm layer thickness), mica 3D components were fabricated using DLP-based AM technology. The original 3D models and the AM-fabricated green, debound, and sintered bodies at 1000 °C are shown in Figure 9. The linear shrinkage after sintering at 1000 °C was 8.62%. A ceramic green body should be printed after enlarging its size by considering the sintering shrinkage rate. Three-dimensional complex mica structures with complex internal geometric pore structures needed for filters or absorbents, which cannot be manufactured by conventional ceramic techniques, can be perfectly fabricated by DLP-based AM without any noticeable cracks, delamination, or defects.

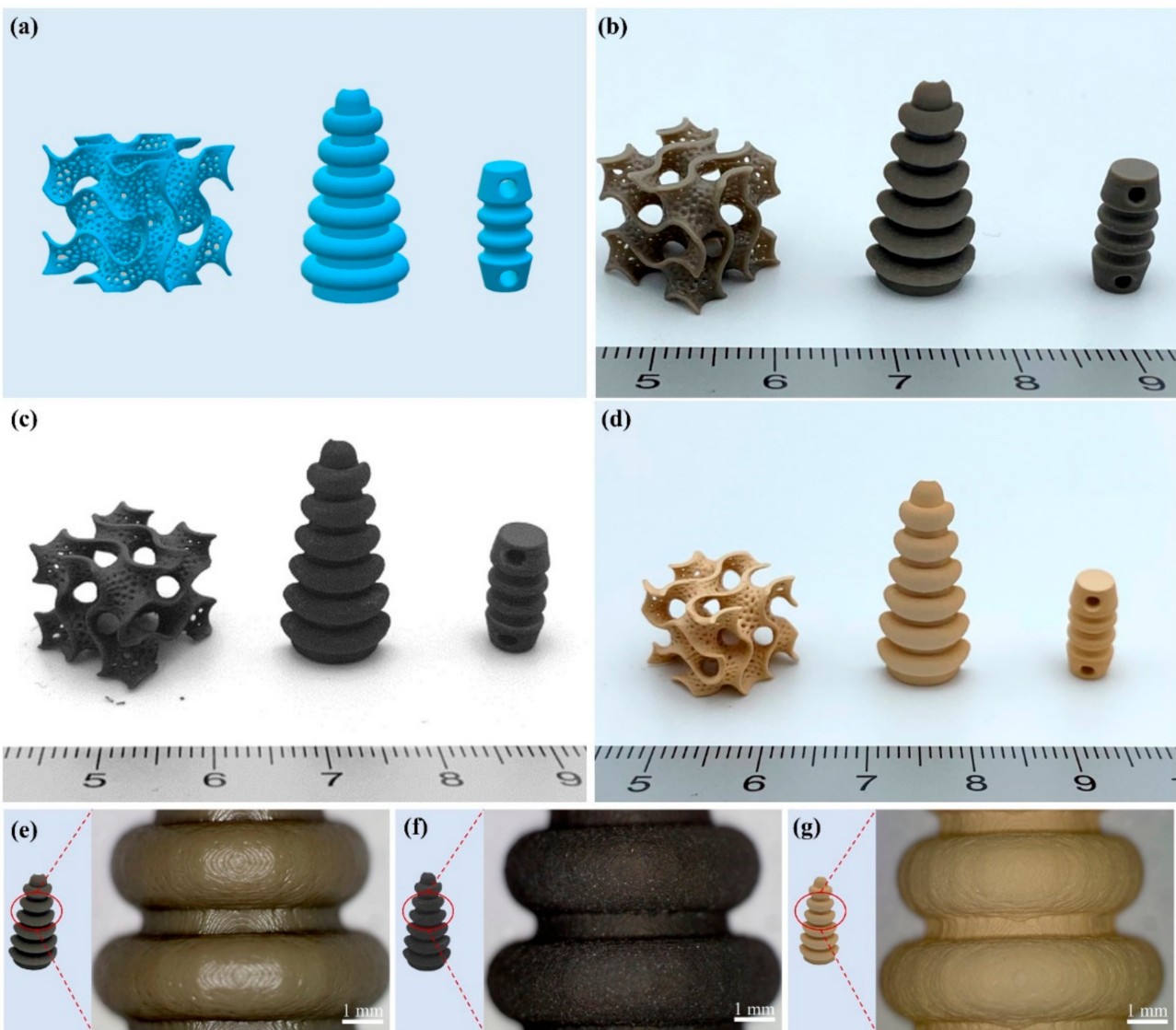

**Figure 9.** (**a**) Three-dimensional models and complex 3D mica structures of the (**b**) green body, (**c**) debound body, (**d**) sintered body at 1000 °C, (**e**) optical image of the green body, (**f**) optical image of debound body, and (**g**) optical image of the sintered body at 1000 °C.

Since the layered structure of mica provides it with excellent adsorption properties, the porous gyroid structure was designed to increase the surface area through a geometrically complex 3D structure, which can potentially be used as a catalyst and a metal filter to improve the adsorption capacity. In addition, mica possesses exceptional insulating characteristics. In order to expand the application scenarios of its insulating properties, additional designs for insulators were successfully fabricated. It has been proven that AM technology can be promising in expanding the ceramic market through the freedom of shape control.

### 4. Conclusions

Mica ceramics with precise, complex geometric shapes have been fabricated using DLP-based AM technology, in which both the slurry preparation and printing parameters are optimized. Planetary milling was used to reduce the particle size and width distribution of the raw mica powder and effectively improve the sedimentation of the slurry. The optimal dispersant concentration of 7 wt.% allows for the most homogenous dispersion of the powder in the slurry. Based on this premise, the solid loading of the mica slurry was increased to 53 vol.% and maintained shear thinning behavior in the low shear rate range, which is favorable for the ceramic DLP process. By evaluating the cure behavior, the layer thickness and exposure energy dose were set to 20 μm and 10 mJ/cm$^2$, respectively, which is sufficient to ensure a high resolution for the final products. Complex-shaped defect-free specimens can be produced after debinding and sintering. This can be significant for expanding the applications of diverse clay minerals, including mica.

**Author Contributions:** Conceptualization, S.Z.; methodology, S.Z., I.A.S., and H.-s.Y.; validation, H.-s.Y.; formal analysis, S.Z., and I.A.S.; investigation, S.Z., I.A.S., and J.K.; writing—original draft preparation, S.Z.; writing—review and editing, S.Z., J.K., Y.-J.C., C.W.G., and H.-s.Y.; visualization, S.Z., J.K., and Y.-J.C.; supervision, H.-s.Y.; project administration, C.W.G. and H.-s.Y. All authors have read and agreed to the published version of the manuscript.

**Funding:** This study was supported by the Fundamental Research Program of the Korea Institute of Materials Science (PNK8170).

**Institutional Review Board Statement:** Not applicable.

**Informed Consent Statement:** Not applicable.

**Data Availability Statement:** Not applicable.

**Conflicts of Interest:** The authors declare no conflict of interest.

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
