# Peer review of "Fabrication of Complex Three-Dimensional Structures of Mica through Digital Light Processing-Based Additive Manufacturing"

_ceramics, doi:10.3390/ceramics5030042_

Round 1

Reviewer 1 Report

1. Describe succinctly what are the processing additives TMPTA, HDDA and the dispersant DISPERBYK in the Methodology section. Explain the reasons for your choosing the additive formulation and the procedure used for preparation of photosensitive resin;

2. It is important to explain the debinding process in detail, temperature rising in debinding is so quick (1C/min), usually in debinding process It is suggested to raise the temperature slowly (around 0.1C/min), fast raising temperature in debinding will lead to crack formation in the sample. Authors are strongly suggested to modify the debinding process.

3. Please mention the soaking time in sintering process.

4. Please explain about density and shrinkage measuring, what kind of sample shape did you use to measure the shrinkage. And how many sample did you measure?

5. It would be interesting if authors could add optical microscope image of the Mica sample after printing debinding and sintering. That can help readers to understand the quality of the printing.

6. It would be great if authors could provide the mechanical properties (such as hardness and compressive strength) of the printed sample and compare the result with mechanical properties of the conventional fabricated Mica.

7. Please explain about XRD result that presented in fig 8, especially about XRD pattern of the sample sintered at 1100 C.

8. Please measure grain size of Mica after different sintering process.

Author Response

The authors have carefully read the comments from the esteemed reviewer, all of which were valuable. We have responded to all comments from the reviewer and have made all the required revisions. We feel that doing so has substantially improved our manuscript. Our answers to the comments and revisions made in the manuscript are given below. For the complete document and Figure, please see the attachment.

Point 1: Describe succinctly what are the processing additives TMPTA, HDDA, and the dispersant DISPERBYK in the Methodology section. Explain the reasons for your choosing the additive formulation and the procedure used for the preparation of photosensitive resin;

Response 1: 1,6-hexanediol diacrylate (HDDA) is widely utilized as a diluent in UV-curable applications due to its low viscosity [1], allowing for the preparation of slurries with higher solid loading. As mentioned in the manuscript, slurry with a higher solid loading is preferable for ceramics DLP printing because it not only reduces the shrinkage rate after debinding and sintering, thereby decreasing the likelihood of cracking and defects but also increases the density of green bodies and sintered samples. However, HDDA is a bifunctional acrylate monomer; therefore, if only HDDA is employed as a resin, the ceramic slurry would have a limited curing ability, which is not conducive to further 3D printing. Therefore, trimethylolpropane triacrylate (TMPTA), a trifunctional acrylate monomer with a stronger cross-linking ability [2], was selected to synergize with HDDA in order to improve the curing ability of the ceramic slurry while maintaining a high solid loading.

As the dispersant DISPERBYK-180, it is alkylolammonium salt of a copolymer with acidic groups, which is commonly used as dispersing additive for stabilizing inorganic particles [3]. We have added a short description regarding the monomers and dispersant in the Materials and Methods section.   

Manuscript revision

(Paragraph 1 of 2. Materials and Methods)

‘Mica (INT Korea, Korea) was utilized as a ceramic powder. DISPERBYK-180 (Altana AG, Germany), a frequently used dispersing additive for stabilizing inorganic particles, was selected as the dispersing agent for producing the slurry. The photosensitive resin consisted of the combination of a trifunctional acrylate monomer, trimethylolpropane triacrylate (TMPTA, Sigma-Aldrich, USA), and a low viscosity bifunctional acrylate monomer, 1,6-hexanediol diacrylate (HDDA, Sigma-Aldrich, USA).’

Point 2: It is important to explain the debinding process in detail, temperature rising in debinding is so quick (1C/min), usually in debinding process It is suggested to raise the temperature slowly (around 0.1C/min), fast raising temperature in debinding will lead to crack formation in the sample. Authors are strongly suggested to modify the debinding process.

Response 2: Thank you for your critique. We quite agree that if the heating rate during debinding is too fast, cracks are likely to form. Nonetheless, the situation will vary depending on the properties of the powder and resin. In general, micron-sized ceramic powders are less susceptible to cracking during debinding than nano-sized ceramic powders. This is because the particle gap between the micron powders is larger, which is more conducive to the volatilization of organic substances during the debinding process. Since the mica powder used in this study has a relatively larger particle size, in order to shorten the experiment time and save energy, we chose a faster heating rate. The results indicated that 1 °C/min was suitable for the debinding of mica, as it did not result in cracks and defects.

Point 3: Please mention the soaking time in sintering process.

Response 3: For the sintering process, based on the TGA results in Figure 1, we assumed that no more reactions would occur until 600 ℃, so the heating rate was set to 5 ℃/min. The rate was then reduced to 1 °C/min to guarantee temperature homogeneity, improved particle orientation, and contact growth. Once the temperature reached the sintering temperature, a 4 h dwell time was set. We have added relevant descriptions in the Materials and Methods section.

Manuscript revision

(Paragraph 4 of 2. Materials and Methods)

‘The specimens were first heated to 600 °C under flowing N2 to remove any remaining organic materials, then they were consolidated by heating at 800, 900, 1000, and 1100 °C for 4 h, respectively, in a box furnace under air.’

Point 4: Please explain about density and shrinkage measuring, what kind of sample shape did you use to measure the shrinkage. And how many sample did you measure?

Response 4: Density and shrinkage measurements used cubic samples with length × width × height of 10 × 5 × 2.5 mm, and the number of samples measured in each condition was 5. We have added size specifications for density and shrinkage measurements in the Materials and Methods section.

Manuscript revision

(Paragraph 7 of 2. Materials and Methods)

‘The density and shrinkage rate of sintered samples were measured using 10 mm × 5 mm × 2.5 mm (length × width × height) cube (n=5).’

Point 5: It would be interesting if authors could add optical microscope image of the Mica sample after printing debinding and sintering. That can help readers to understand the quality of the printing.

Response 5: Thank you for your suggestion. The optical microscope images were added.

Manuscript revision

(Paragraph 1 of 3.4. Fabrication of a 3D complex structure of mica)

‘Figure 9. (a) 3D models and mica complex 3D structures of the (b) green body, (c) debound body, (d) sintered body at 1000 °C, (e) optical image of green body, (f) optical image of debound body and (g) optical image of sintered body at 1000 °C.’

Point 6: It would be great if authors could provide the mechanical properties (such as hardness and compressive strength) of the printed sample and compare the result with mechanical properties of the conventional fabricated Mica.

Response 6: Thank you for your suggestion. Since our research is mainly focused on exploring the possibility of using mica for DLP 3D printing, we did not further explore its mechanical properties. Certainly, in future work, we will conduct specific structural design and research on various properties according to different application scenarios of mica.

Point 7: Please explain about XRD result that presented in fig 8, especially about XRD pattern of the sample sintered at 1100 C.

Response 7: Figure 8 demonstrates that when the sintering temperature was less than 1100 °C, the XRD patterns were consistent with those of potassium mica. This was due to the dehydroxylation reaction of the raw powders caused by high-temperature sintering, which is a characteristic phenomenon of muscovite at high temperatures [4]. When the sintering temperature reached 1100 °C, it can be seen that many characteristic peaks of mica disappeared, and there were only a few small peaks. This might be due to the fact that 1100 °C has reached the melting point of the mica raw material used in the experiment, destroying its crystal structure. Mica should have recrystallized during the furnace cooling, as evidenced by the presence of a few tiny peaks on the XRD spectrum. We have improved explanation of XRD result in the section 3.4.

Manuscript revision

(Paragraph 2 of 3.4. Properties of the structure of mica ceramics)

‘When the temperature was below 1100 °C, mica retained its lattice structure; however, the main characteristic peaks disappeared after the temperature of 1100 °C was reached, indicating that the crystal structure was destroyed. This might be because the mica powder used in this experiment has reached its melting point at 1100 °C, and after sintering, the mica may recrystallize during the furnace cooling causing minor peaks to appear in the XRD spectrum. This observation is consistent with the results of FE-SEM (Figure 7).’

Point 8: Please measure grain size of Mica after different sintering process.

Response 8: Thank you for your suggestion. Different from ordinary ceramics, for clay minerals, their mineralogy and excellent adsorption properties are mainly derived from the crystal structure [5]. As we mentioned in the manuscript, mica has a layered structure with two layers of silicic acid tetrahedron sandwiching one layer of an aluminum–oxygen octahedron. The isomorphous substitution of Si/Al/Mg in the tetrahedron/octahedron by low-valent cations imparts a permanent negative charge to mica, making it an effective adsorbent for the treatment of cationic pollutants. Consequently, it is essential to preserve the features of mica crystals after heat treatment. Thus, after sintering, mica mainly maintains the shape and size of the raw particles, so we think it is not necessary to measure the grain size again.

References:

  1. Nabeth, B.; Gerard, J. F.; Pascault, J. P. Dynamic Mechanical Properties of UV‐curable Polyurethane Acrylate with Various Reactive Diluents. Journal of Applied Polymer Science. 1996, 60, 2113–2123.
  2. Nason, C.; Roper, T.; Hoyle, C.; Pojman, J. A. UV-Induced Frontal Polymerization of Multifunctional (Meth)Acrylates. Macromolecules. 2005, 38, 5506–5512.
  3. Kim, J.-H.; Maeng, W.-Y.; Koh, Y.-H.; Kim, H.-E. Digital Light Processing of Zirconia Prostheses with High Strength and Translucency for Dental Applications. Ceramics International. 2020, 46, 28211–28218.
  4. Keppler, H. Ion Exchange Reactions between Dehydroxylated Micas and Salt Melts and the Crystal Chemistry of the Interlayer Cation in Micas. American Mineralogist. 1990, 75, 529–538.
  5. Martín, J.; Orta, M. del M.; Medina-Carrasco, S.; Santos, J. L.; Aparicio, I.; Alonso, E. Evaluation of a Modified Mica and Montmorillonite for the Adsorption of Ibuprofen from Aqueous Media. Applied Clay Science. 2019, 171, 29–37.

Reviewer 2 Report

In the submitted manuscript, Mica structures could be fabricated successfully by an additive manufacturing technique of digital light processing though systematic optimizations of particles shape and dispersions in liquid slurries. As a reviewer, I think it is better for the author to describe clearly about practical applications of fabricated components.  Of course, part accuracies are  sophisticated enough through researches between irradiation powers and curing deps with light scattering. The created objects shown in Fig. 9 should be explain if they have enough part accuracy and geometric structures in the considerations of practical applications.

Author Response

The authors have carefully read the comments from the esteemed reviewer, all of which were valuable. We have responded to all comments from the reviewer and have made all the required revisions. We feel that doing so has substantially improved our manuscript. Our answers to the comments and revisions made in the manuscript are given below.

Point 1: In the submitted manuscript, Mica structures could be fabricated successfully by an additive manufacturing technique of digital light processing though systematic optimizations of particles shape and dispersions in liquid slurries. As a reviewer, I think it is better for the author to describe clearly about practical applications of fabricated components.  Of course, part accuracies are  sophisticated enough through researches between irradiation powers and curing deps with light scattering. The created objects shown in Fig. 9 should be explain if they have enough part accuracy and geometric structures in the considerations of practical applications.

Response 1: Thank you for your suggestions. In this study, we mainly demonstrate the possibility of using DLP 3D printing technology to control the 3D structure of mica to broaden its application. In the last part of the manuscript, we present various 3D samples designed and printed based on the application of mica. Based on the adsorption properties of mica, the porous gyroid structure in Figure 9 was created to function as a catalyst or a filter. The geometrically complicated 3D porous structure can significantly increase the surface area of mica, hence enhancing its absorption capacity. The other two constructs are insulator structures printed based on the insulating properties of mica to demonstrate the possibilities of 3D printing in this regard. Since this manuscript mainly focuses on the optimization of slurry and printing parameters to realize the possibility of DLP 3D printing mica. Consequently, future studies will focus on discussing and analyzing the performance of these structures when employed in practical situations.

Manuscript revision

(Paragraph 2  of 3.5. Fabrication of a 3D complex structure of mica)

‘Since the layered structure of mica provides it with excellent adsorption properties, the porous gyroid structure was designed to increase the surface area through a geometrically complex 3D structure, which can potentially be used as a catalyst and a metal filter to improve the adsorption capacity. In addition, mica possesses exceptional insulating characteristics. In order to expand the application scenarios of its insulating properties, additional designs for insulators were successfully fabricated. It has been proven that AM technology can be promising in expanding the ceramic market through the freedom of shape control.’
